# Genetic Determinants of *Acinetobacter baumannii* Serum-Associated Adaptive Efflux-Mediated Antibiotic Resistance

**DOI:** 10.3390/antibiotics12071173

**Published:** 2023-07-11

**Authors:** Mikaeel Young, Michaelle Chojnacki, Catlyn Blanchard, Xufeng Cao, William L. Johnson, Daniel Flaherty, Paul M. Dunman

**Affiliations:** 1Department of Microbiology and Immunology, University of Rochester Medical Center, Rochester, NY 14642, USA; mikaeel_young@baylor.edu (M.Y.); mchojnacki@trudeauinstitute.org (M.C.); wl.johnsonpubhealth@gmail.com (W.L.J.); 2Department of Medicinal Chemistry and Molecular Pharmacology, Purdue University College of Pharmacy, Lafayette, IN 47907, USA; 3Purdue Institute for Drug Discovery, West Lafayette, IN 47907, USA; 4Purdue Institute of Inflammation, Immunology and Infectious Disease, West Lafayette, IN 47907, USA

**Keywords:** *Acinetobacter baumannii*, adaptive efflux, antibiotic tolerance, serum

## Abstract

*Acinetobacter baumannii* is a nosocomial pathogen of serious healthcare concern that is becoming increasingly difficult to treat due to antibiotic treatment failure. Recent studies have revealed that clinically defined antibiotic-susceptible strains upregulate the expression of a repertoire of putative drug efflux pumps during their growth under biologically relevant conditions, e.g., in human serum, resulting in efflux-associated resistance to physiologically achievable antibiotic levels within a patient. This phenomenon, termed Adaptive Efflux Mediated Resistance (AEMR), has been hypothesized to account for one mechanism by which antibiotic-susceptible *A. baumannii* fails to respond to antibiotic treatment. In the current study, we sought to identify genetic determinants that contribute to *A. baumannii* serum-associated AEMR by screening a transposon mutant library for members that display a loss of the AEMR phenotype. Results revealed that mutation of a putative pirin-like protein, YhaK, results in a loss of AEMR, a phenotype that could be complemented by a wild-type copy of the *yhaK* gene and was verified in a second strain background. Ethidium bromide efflux assays confirmed that the loss of AEMR phenotype due to pirin-like protein mutation correlated with reduced overarching efflux capacity. Further, flow cytometry and confocal microscopy measures of a fluorophore 7-(dimethylamino)-coumarin-4-acetic acid (DMACA)-tagged levofloxacin isomer, ofloxacin, further verified that YhaK mutation reduces AEMR-mediated antibiotic efflux. RNA-sequencing studies revealed that YhaK may be required for the expression of multiple efflux-associated systems, including MATE and ABC families of efflux pumps. Collectively, the data indicate that the *A. baumannii* YhaK pirin-like protein plays a role in modulating the organism’s adaptive efflux-mediated resistance phenotype.

## 1. Importance

Intrinsic *A. baumannii* antibiotic resistance is predominantly attributable to mutations that lead to overexpression of one or more of the organism’s 40 putative efflux pump systems [1]. In addition to intrinsic resistance, it is well recognized that bacteria elicit adaptive resistance mechanisms during infection that allow them to resist physiologically achievable antibiotic concentrations within a patient (reviewed in [2]). In that regard, laboratory-defined antibiotic-susceptible *A. baumannii* has the remarkable ability to induce an expansive repertoire of drug efflux pumps during its growth under physiological salt conditions, e.g., in pulmonary surfactant or human serum, thereby allowing them to become resistant to physiologically achievable antibiotic concentrations within a patient [3,4,5,6,7]. This phenomenon, termed adaptive efflux-mediated antibiotic resistance (AEMR), is conserved across *A. baumannii* lineages as well as other bacterial species, such as *Pseudomonas aeruginosa* [8]. The current study was designed to define genetic determinants that modulate *A. baumannii* AEMR, as such factors may represent targets for the therapeutic development of agents that reduce antibiotic resistance.

## 2. Introduction

Antibiotic-resistant bacteria accounted for 2.8 million infections and 35,000 deaths in the U.S. in 2019 and are predicted to cause 10 million infections and 700,000 deaths globally by the year 2050 [9,10]. In that regard, *A. baumannii* has emerged as a particularly problematic pathogen. Indeed, the latest National Healthcare Safety Network (NHSN) report revealed that *A. baumannii* accounted for approximately 3% of all nosocomial infections, 7.8% of ventilator-associated pneumonias, and 2.2% of all hospital central line-associated bloodstream infections in the U.S. in 2006 [11]. Simultaneously *A. baumannii* antibiotic resistance has escalated at a rapid pace; 47.7% of clinical isolates collected from hospitalized patients in the U.S. between 2013 to 2017 were found to be multidrug-resistant, and a staggering 37.5% were not susceptible to carbapenems, which have been considered front-line therapeutics for severe *A. baumannii* infections [12]. In response, the World Health Organization, Centers for Disease Control and Prevention, and Infectious Diseases Society of America have all recognized *A. baumannii* as a pathogen of urgent healthcare concern, and the organism has been designated as one of the six ESKAPE (*Enterococcus faecium*, *Staphylococcus aureus*, *Klebsiella pneumoniae*, *A. baumannii*, *P. aeruginosa*, and *Enterobacter* sp.) bacterial pathogens of greatest concern that can ‘escape’ the effects of antibiotics due to resistance [9,13,14].

*A. baumannii* antibiotic resistance is, in part, attributable to the presence of antibiotic-modifying enzymes but is generally regarded to be a consequence of the organism’s expansive repertoire of antibiotic efflux pumps. *A. baumannii* is capable of producing each of the six major classes of drug efflux pumps, including the resistance nodulation division (RND) pumps AdeABC, AdeFGH, and AdeIJK, the major facilitator superfamily (MFS) pumps CraA and AmvA, the multidrug and toxic compound extrusion (MATE) pump AbeM, small multidrug resistance (SMR) AbeS pump, proteobacterial antimicrobial compound efflux (PACE) pump, AceI, and the ATP-binding cassette (ABC) family of pump MacB, which collectively confer resistance to virtually all front-line antibiotics (reviewed in [1]). While a multitude of *A. baumannii* antibiotic efflux pumps have been characterized, comparative genomics indicates that the genomes of strains ATCC 17978 and/or AYE contain more than 45 putative additional drug efflux-like transporters, any one of which may also contribute to antibiotic resistance [8,15].

Antibiotic drug efflux pumps are tightly regulated, and most are repressed or only basally expressed during conventional laboratory growth conditions, requiring mutations that lead to their constitutive expression to achieve an intrinsic antibiotic resistance phenotype, but often at a significant fitness cost [16]. It is becoming increasingly appreciated that while such efflux pumps may not be expressed at a high level in ordinary laboratory conditions, they can be induced in response to stress conditions that an organism encounters during infection. Such induction may allow a seemingly laboratory-defined antibiotic-susceptible bacterial strain to tolerate or resist antibiotic treatment. Indeed, while *Escherichia coli* strain K12 codes for at least 36 efflux pumps, only one, AcrAB-TolC, is appreciably expressed during exponential growth in nutrient-rich media at 37 °C [17,18]. Conversely, under starvation conditions, *E. coli* induces at least 10 efflux systems that modulate resistance to the antibiotic ampicillin, whereas anaerobic growth and low pH induce expression of the RND-like efflux pump, MdtEF, and lead to antibiotic resistance [18,19,20]. Conditions that mimic the infection setting have also been shown to induce efflux systems in other bacterial species. For instance, *Mycobacterium tuberculosis*, *Bacillus cereus*, and *Salmonella enterica* induce efflux pumps that modulate antibiotic resistance in response to macrophage internalization, oxidative stress, and bile exposure, respectively [21,22,23].

Similarly, recent studies have revealed that laboratory-defined antibiotic-susceptible *A. baumannii* strains upregulate a multitude of efflux pumps in response to biologically relevant host conditions, such as physiological salt concentrations and human serum, and display efflux-mediated resistance to physiologically achievable concentrations of aminoglycosides, carbapenems, fluoroquinolones, tetracyclines, and the glycylcycline antibiotic, tigecycline, within a patient [7,8]. This phenomenon, termed adaptive efflux-mediated resistance (AEMR), has been hypothesized to account for one mechanism by which antibiotic-susceptible strains-resist antibiotic treatment within the host setting, ultimately leading to clinical antibiotic failure [24]. While serum-associated AEMR has been observed in *A. baumannii* as well as *P. aeruginosa* clinical isolates, the underlying genetic determinants that modulate this phenomenon are poorly understood [8,25]. Accordingly, the objective of the current study was to identify genetic determinants that contribute to *A. baumannii* AEMR. To do so, an *A. baumannii* transposon mutant library was screened for members that display a loss of serum-associated efflux-mediated antibiotic resistance phenotype. Complementation and efflux assays coupled with confocal microscopy, flow cytometry, and RNA sequencing (RNA-seq) analyses collectively revealed that the *A. baumannii* YhaK pirin-like protein (ATCC 17978 A1S_3277) is likely to be a key component of the organism’s AEMR to the antibiotics minocycline and levofloxacin.

## 3. Materials and Methods

### 3.1. Bacterial Strains and Growth Conditions

The bacterial strains and plasmids used in this study are listed in Table 1. *A. baumannii* strain 98-37-09 is a well-characterized antibiotic-susceptible cerebrospinal fluid clinical isolate obtained from the Centers for Disease Control and Prevention [8,26,27,28]. The *A. baumannii* strain 98-37-09 EZ-Tn5 transposon mutant library used in these studies is comprised of 6000 members that have been individually arrayed in a 96-well plate format and have been previously described [28]. *A. baumannii* strain 98-37-09-YhaK is an EZ-Tn5-mutant library member that harbors a transposon insertion within the strain’s YhaK gene (A1S_3277:Tn5), whereas strains 98-37-09-YhaK pWH1266 and 98-37-09-YhaK pYhaK are corresponding mutants that have been transformed with the plasmid shuttle-vector pWH1266 [29] or pWH1266 harboring a wild-type copy of the YhaK gene (pYhaK), respectively, and were created for this study as described below. *A. baumannii* strain AB5075-UW is a clinical isolate obtained from a combat wound infection, whereas strain AB5075-YhaK is an AB5075-UW derivative containing a transposon insertion with the YhaK locus (ABUW_0207:T26) and were obtained from the Manoil *A. baumannii* mutant library [30] from the University of Washington (Seattle, WA, USA). *A. baumannii* strains AB5075-YhaK pWH1266 and AB5075-YhaK pYhaK are AB5075-YhaK derivatives containing plasmid pWH1266 [29] and pYhaK, respectively, and created for this study, as described below. Chemically competent OneShot^®^ TOP10 *Escherichia coli* were used for cloning and plasmid construction (Thermo Fisher Scientific, Waltham, MA, USA). All *A. baumannii* and *E. coli* strains were grown in Luria-Bertani (LB) broth and/or 100% human serum (Corning Life Sciences, Tewksbury, MA, USA) at 37 °C in media supplemented with 50 µg mL^−1^ of kanamycin or 10 µg mL^−1^ tetracycline, as indicated.

### 3.2. Transposon Mutant Screening for Loss of Adaptive Efflux-Associated Resistance

Each member of an *A. baumannii* strain 98-37-09 transposon mutant library was individually grown overnight in independent wells of a microtiter plate containing 100 µL LB broth supplemented with kanamycin at 37 °C. For screening, mutants were transferred to individual wells of a new microtiter plate (diluted 1:100) containing 100 µL fresh medium and grown to optical density 600_nm_ (OD600_nm_) of approximately 0.4–0.5 at 37 °C. Approximately 1 × 10^5^ colony forming units (CFU) of each transposon mutant library member were then transferred to individual wells of a 96-well round bottom plate containing 100 µL of 100% human serum supplemented with 1.0 µg mL^−1^ minocycline (0.5× MIC in human serum). Plates were incubated at 37 °C for 48 h, and bacterial growth was detected by the unaided human eye. Transposon library members that displayed a loss of growth phenotype were expected to include mutants that exhibited a loss of efflux-associated antibiotic tolerance and were selected for further study.

### 3.3. Identification of Transposon Insertion Site of Mutants of Interest

Inverse PCR was used to identify the transposon insertion sites of transposon library members of interest, as previously described [28]. Briefly, total bacterial DNA was purified from each mutant using DNeasy Blood and Tissue kits (Qiagen, Valencia, CA, USA). Two micrograms of DNA was digested with the restriction enzyme *Afe*I (10 U; New England Biolabs, Ipswich, MA, USA) for 2 h at 37 °C and heat-inactivated at 65 °C for 5 min. Restriction fragments were circularized by ligation using 1.5 units of T4 DNA ligase (Thermo Fisher, Waltham, MA, USA) for 16 h at 16 °C. Following ligase heat-inactivation, PCR was performed using Platinum PCR Supermix High Fidelity Kits (Invitrogen, Waltham, MA, USA) and transposon-specific primers (forward, 5′-ACCTACAACAAAGCTCTCATCAACC-3′; reverse, 5′-CTACCCTGTGGAACACCTACATCT-3′) in a GeneAmp PCR System 9700 thermocycler (Applied Biosystems, Foster City, CA, USA) with the following parameters: 94 °C for 10 min and 50 cycles of 94 °C for 30 s, 57 °C for 30 s, and 72 °C for 6 min, followed by an extension at 72 °C for 10 min. PCR products were electrophoresed in a 1% UltraPure Agarose gel (Invitrogen) at 75 V for 40 min and gel-purified using QIAquick Gel Extraction Kits (Qiagen). Purified products were cloned into plasmid pCR2.1-TOPO using TOPO TA cloning kits and then transformed into 50 μL TOP10 *E. coli*, following the manufacturer’s recommendations (Thermo Fisher, Waltham, MA, USA). Transformants were selected on LB agar supplemented with 50 μg mL^−1^ kanamycin and screened for plasmid inserts by colony PCR. To do so, approximately half of each colony was combined with DreamTaq Green PCR Mastermix (Thermo Fisher, Waltham, MA, USA) and amplified using plasmid-directed M13 sequencing primers (Forward: 5′-GTAAAACGACGGCCAG-3′, Reverse 5′-CAGGAAACAGCTATGAC-3′) using the following parameters: 95 °C for 5 min and 35 cycles of 95 °C for 1 min, 54 °C for 1.5 min, and 72 °C for 5 min, followed by a final extension at 72 °C for 5 min. PCR products were electrophoresed in a 1% UltraPure Agarose gel. Colonies containing a plasmid insert were propagated, and plasmid DNA was isolated using the Qiaprep Spin Miniprep kit (Qiagen) and then sequenced by ACGT Incorporated (Wheeling, IL, USA) using M13 sequencing primers. Sequences were aligned utilizing BLAST against the *A. baumannii* strain 17978 (NCBI:txid400667) genome to map the transposon insertion site [31].

### 3.4. Adaptive Antibiotic Tolerance Assays

*A. baumannii* strains 98-37-09, AB5075, and corresponding mutants of interest were propagated overnight in LB broth at 37 °C, diluted into fresh medium (1:100), and grown to OD600_nm_ 0.4–0.5 at 37 °C with aeration. A total of approximately 1 × 10^5^ cells were transferred to individual wells of a 96-well round bottom plate (Corning Life Sciences, Corning, NY, USA) containing 100 μL of LB (negative control) or 100% human serum (AEMR inducing condition) supplemented with 0, 0.125, 0.25, 0.5, 1, 2, or 4.0 µg mL^−1^ minocycline, levofloxacin, ofloxacin, or with 0, 25, 50, 100, or 200 µg mL^−1^ DMACA-labeled ofloxacin (described below) and incubated at 37 °C for 48 h. To quantify the antimicrobial effects of each antibiotic, cells were serially diluted in 0.8% sodium chloride and plated on LB agar to enumerate the viable number of CFU mL^−1^ for each strain during growth in LB and serum supplemented with various concentrations of the indicated antibiotic. All experiments were performed at least 3 times, and results were averaged and presented with ±standard deviation.

### 3.5. Ethidium Bromide Efflux Assays

Bacterial ethidium bromide (EtBr) efflux assays were performed to assess the ability of the indicated *A. baumannii* strain to efflux EtBr, as previously described [26]. *A. baumannii* strains were propagated for 16 h in LB broth at 37 °C with aeration, diluted 1:100 in fresh LB (negative control) or 100% human serum, and grown to the mid-exponential phase. Cells were collected by centrifugation at 900× *g* for 20 min at 4 °C, washed three times with sodium phosphate buffer (15 mM Na_2_HPO_4_ and 5 mM NaH_2_PO_4_), and resuspended to a final OD600_nm_ of 0.2 in sodium phosphate buffer. A total of 1 × 10^6^ CFUs were transferred to individual wells of a 96-well white plate (Falcon, Corning Life Sciences), containing either dimethyl sulfoxide (DMSO; negative control) or 40 μg mL^−1^ of the efflux pump inhibitor phenylalanine arginine β-naphthylamide (PAβN; MP Biomedicals, Irvine, CA, USA). Ethidium bromide (10 μg mL^−1^) was added, and corresponding fluorescence was monitored (excitation, 530_nm_; emission, 600_nm_) every 5 min for a total of 90 min on a SPECTRAmax5 fluorimeter (Molecular Devices, Sunnyvale, CA, USA) and averaged. All experiments were performed at least 3 times, and results were averaged and presented with ± standard deviation.

### 3.6. Outer Membrane Integrity Assay

Bacterial cell membrane permeability measures were performed using 1-*N*-phenylnapthylamine (NPN) assays in triplicate, as previously described [32]. Briefly, *A. baumannii* strains were propagated for 16 h in LB broth at 37 °C with aeration, diluted 1:50 in fresh human serum, and grown to the mid-exponential phase. Cells were harvested by centrifugation (3000× *g*), washed twice with assay buffer (5 mM HEPES, 5 mM glucose, pH 7.2), and resuspended in assay buffer to a final OD600_nm_ = 1.0. In triplicate, a total of 100 μL of washed cells were mixed with an equal volume of assay buffer containing 20 μM NPN and placed into individual wells of a 96-well black plate (Corning Life Sciences, Corning, NY, USA); A volume of 5 µL of assay buffer or 0.06% Triton X-100 served as negative and positive controls, respectively. Fluorescence was measured at an excitation wavelength of 350_nm_ and emission wavelength of 420_nm_, averaged, and normalized to wild-type cells. All experiments were performed at least 3 times, and results were averaged and presented with ±standard deviation. 

### 3.7. Construction of Complementation Plasmids

The *A. baumannii* 17978 YhaK locus A1S_3277 and 500 base pair upstream sequence were cloned from strain 98-37-09 and placed into the multiple cloning site of the shuttle vector pWH1266 [29]. To do so, 98-37-09 genomic DNA served as a template for PCR amplification using A1S_3277 forward primer 5′-ATGCCTGCAGAGTTGTTTTTTACCAGTTGCTTTAACCGC-3′ and reverse primer 5′-ATGCCTGCAGTTATCCTTCTTGCATGATTGAACCGAATTTTCC-3′, with each oligonucleotide containing a PstI restriction enzyme site (underlined) using DreamTaq PCR Mastermix Kits (Thermo Fisher). Resulting PCR products were separated by 0.8% agarose gel electrophoresis, purified using QIAquick Gel Extraction Kits, ligated into plasmid pCR2.1-TOPO using TOPO TA cloning kits, and transformed into *E. coli* TOP10 cells following the manufacturer’s recommendations (Thermo Fisher, Waltham, MA, USA). Transformants were selected by growth on LB agar supplemented with 50 μg mL^−1^ tetracycline and propagated, and plasmid DNA was isolated using Qiaprep Spin Miniprep kits (Qiagen). Purified plasmids were digested with PstI to liberate the plasmid insert, which was gel-purified using QIAquick Gel Extraction Kits and ligated into PstI-digested pWH1266 shuttle vector using T4 DNA ligase (Thermo Fisher) to generate plasmid pYhaK (A1S_3277). Constructs were first transformed into *E. coli* TOP10 for plasmid propagation, purified using Qiaprep Spin Miniprep kits, and sequenced to ensure their integrity (ACGT Incorporated, Wheeling, IL, USA). Plasmids pWH1266 and pYhaK were electroporated into *A. baumannii* 98-37-09-YhaK and AB5075-YhaK and selected with 50 μg mL^−1^ tetracycline for complementation studies.

### 3.8. Real-Time Quantitative PCR to Measure A1S_3277 Expression

In triplicate, *A. baumannii* strains were grown to exponential phase in either LB media or human serum in the absence or presence of 0.15 µg mL^−1^ levofloxacin. Total bacterial RNA was isolated using Qiagen RNeasy kits, DNase treated, repurified and used as a template (400 ng) for qRT-PCR using primers for A1S_3277 (forward, 5′-TCGATGAAGAAATTTCTTGGTGCATATCAAAATAACC-3′; reverse, 5′-ATGCCTGCAGTTATCCTTC TTGCATGATTGAACCGAATTTTCC-3′) and as a template (4 ng) for 16S rRNA (forward,5′-CTGTAGCGGGTCTGAGAGGAT-3′; reverse, 5′-CCATAAGCCTTCTTCACAC-3′) amplified, and measured using PerfeCta SYBR Green Fastmix and qScript cDNA SuperMix kits following the manufacturer’s recommendations (QuantaBio, Beverly, MA, USA). All samples were conducted in triplicate and normalized to 16S rRNA, averaged, and compared to untreated exponential phase wild-type cells grown in LB media.

### 3.9. Synthesis of DMACA-Ofloxacin

In the first step, 9-Fluoro-3-methyl-7-oxo-10-(piperazin-1-yl)-2,3-dihydro-7*H*-[1,4]oxazino[2,3,4-*ij*]quinoline-6-carboxylic acid (Intermediate 1) was synthesized as follows. Under argon atmosphere, piperazine (1.03 g, 12.0 mmol, 3 eq) was added to a suspension of 9,10-difluoro-3-methyl-7-oxo-2,3-dihydro-7*H*-[1,4]oxazino[2,3,4-*ij*]quinoline-6-carboxylic acid (0.56 g, 2.00 mmol, 1 eq) in anhydrous DMSO (10 mL). The suspension was stirred at 95 °C for 16 h. Cold acetone was then added to the mixture, resulting brown precipitate that was triturated in acetone to give intermediate 1 (0.60 g, 1.74 mmol, yield 87%) as a pale-yellow solid. 1H NMR (500 MHz, DMSO-d6) δ 8.93 (s, 1H), 7.53 (d, *J* = 12.4 Hz, 1H), 4.88 (dd, *J* = 13.1, 6.4 Hz, 1H), 4.54 (d, *J* = 11.4 Hz, 1H), 4.38–4.25 (m, 1H), 2.69 (s, 4H), 2.47–2.45 (m, 4H), 1.40 (d, *J* = 6.7 Hz, 3H). ^13^C NMR (126 MHz, DMSO-*d*_6_) *δ* 176.63, 166.46, 156.82, 154.86, 146.48, 140.47, 140.42, 132.70, 125.14, 119.94, 119.86, 109.23, 107.10, 103.72, 103.53, 68.32, 55.10, 51.83, 46.50, 45.45, 18.25. MS (ESI, Advion Mass Spec LC-MS) *m*/*z*: 348.1 (M+1)^+^. In the second step, 10-(4-(2-(7-(dimethylamino)-2-oxo-2H-chromen-4-yl)acetyl)piperazin-1-yl)-9-fluoro-3-methyl-7-oxo-2,3 dihy-dro-7*H*-[1,4]oxazino[2,3,4-*ij*]quinoline-6-carboxylic acid (DMACA-Oflox) was created, as follows. Under an argon atmosphere, a solution was prepared of 2-(7-(dimethylamino)-2-oxo-2*H*-chromen-4-yl)acetic acid (DMACA, 0.22 g, 0.90 mmol, 1.5 eq), HOBt (0.18 g, 1.20 mmol, 2 eq), and (3-dimethylaminopropyl)-ethyl-carbodiimide HCl (0.23 g, 1.20 mmol, 2 eq) in *anhydrous N*,*N*-dimethylformamide (DMF, 8 mL) was stirred in an ice bath, then *N*,*N*-diisopropylethylamine (0.36 mL, 2.40 mmol) was added into the reaction mixture and then stirred at 0 °C for 30 min. A solution of intermediate 1 (0.21 g, 0.60 mmol, 1 eq) in anhydrous DMF (4 mL) was added, and the mixture was heated at 80 °C overnight. The reaction mixture was extracted with ethyl acetate and H_2_O, the organic phase was washed with H_2_O, Brine, then dried with anhydrous Na_2_SO_4_, filtered, and concentrated under vacuum, and the crude product was purified by manual silica gel column Chromatography (0% MeOH/DCM~10% MeOH/DCM) to afford DMACA-Oflox (0.030 g, yield 9%) as a pale yellow solid. ^1^H NMR (500 MHz, CDCl_3_) *δ* 8.63 (s, 1H), 7.75 (d, *J* = 11.8 Hz, 1H), 7.47 (d, *J* = 9.0 Hz, 1H), 6.64 (dd, *J* = 9.0, 2.4 Hz, 1H), 6.52 (d, *J* = 2.4 Hz, 1H), 6.02 (s, 1H), 4.49 (d, *J* = 11.3 Hz, 2H), 4.40–4.33 (m, 1H), 3.84 (s, 4H), 3.60 (s, 2H), 3.35 (t, *J* = 12.5 Hz, 4H), 3.07 (s, 6H), 1.62 (d, *J* = 6.7 Hz, 3H). MS (ESI, Advion Mass Spec LC-MS) *m*/*z*: 576.7 (M+1)^+^.

### 3.10. Flow Cytometry

Flow cytometry was used to measure the cellular accumulation of the levofloxacin isomer, ofloxacin, labeled with the fluorophore 7-(dimethylamino)-coumarin-4-acetic acid (DMACA-Oflox; optimal excitation 390_nm_/emission 480_nm_; MIC 100 µM) in *A. baumannii* strains 98-37-09 and 98-37-09 YhaK cultured in LB or human serum; all conditions were repeated at least three times. To do so, strains were grown overnight in LB and then sub-cultured (1:50 dilution) in either LB or serum for 2.5 h at 37 °C, washed twice in PBS, and then resuspended in PBS to an OD600_nm_ of 0.4. Approximately 2 × 10^7^ cells were returned to 2 mL of fresh LB or serum-supplemented with 75 µM DMACA-Oflox in the presence or absence of 20 µg mL^−1^ *A. baumannii* efflux-pump inhibitor 4-[(4-Methoxybenzyl)oxy] benzene sulfonamide (AE0003; ref. [33]) and incubated for 30 min at 37 °C. Cells were washed and resuspended in PBS and flowed through a BD LSR II flow cytometer (BD Biosciences, Franklin Lakes, NJ, USA). Cells were excited with a violet 405_nm_ laser using a 450/50 bandpass filter. For each sample, fluorescence intensity and side-scatter measures were recorded for a total of 100,000 events and analyzed using Flow Cytometry Standard (FCS) Express (De Novo Software, Pasadena, CA, USA).

### 3.11. Confocal Microscopy

Confocal microscopy was used to measure the cellular association of DMACA-Oflox with *A. baumannii* strains 98-37-09 and 98-37-09 YhaK when cultured in LB or human serum. To do so, strains were grown overnight in LB, sub-cultured (1:50 dilution) in LB or serum for 2.5 h at 37 °C, and then washed and resuspended in PBS to a final OD600_nm_ of 0.4. Approximately 2 × 10^7^ cells were returned to 2 mL fresh LB or serum containing 75 µM DMACA-Olfox in the presence or absence of 20 µg mL^−1^ *A. baumannii* efflux-pump inhibitor AE0003 and incubated for 30 min at 37 °C. Cells were then washed with PBS and fixed with 4% formaldehyde in PBS for 30 min. After fixing, cells were washed in PBS, added to an 8 well-chambered coverglass (Nunc Lab-Tek, Nunc, Rochester, NY, USA), and stored at 4 °C for microscopy using a Nikon A1R confocal microscope, excited with a 405_nm_ laser, and resulting images were analyzed using Imaris 8 software (Oxford Instruments, Abingdon, UK).

### 3.12. RNA Sequencing

RNA sequencing was performed to compare the transcriptomes of *A. baumannii* strains 98-37-09 and 98-37-09 YhaK grown in human serum supplemented with 0.15 µg mL^−1^ levofloxacin. To do so, in duplicate, each strain was grown to the mid-exponential phase in human serum, and total bacterial RNA was isolated from each sample using Qiagen RNeasy kits. Library preparation and Next Generation Sequencing were performed at CD Genomics (Shirley, New York, NY, USA). In brief, RNA concentration and quality were measured using Agilent 2100 Bioanalyzer and normalized, and rRNA was depleted using Ribo-Zero rRNA removal kits and fragmented. To construct sequencing libraries, cDNA synthesis was performed, the 3′ ends were adenylated, and adaptors were ligated. Paired-end sequencing was performed with Illumina NovaSeq PE150 (~20 million reads/sample, Illumina, San Diego, CA, USA). Processed/cleaned reads were then mapped to the *A. baumannii* ATCC 17978 reference genome (ASM1542v1) using Hierarchial Indexing for Spliced Alignment of Transcripts software (HISAT2 version 2.2.1) [34]. Transcripts were quantified, and gene expression levels were determined using Cufflinks software (v 2.2.1). Differentially Expressed Genes (DEGs) were classified based on their Gene Ontology (GO) functional terms and KEGG pathway enrichment analysis using the topGO software (Version 2.8). Raw sequences were deposited at the National Center for Biotechnology Information Sequence Read Archive (NCBI-SRA), under gene expression omnibus accession number GSE190119.

### 3.13. Statistics

Statistical analysis of flow cytometry and confocal microscopy data were performed using Graphpad prism 9, one-way Analysis of Variance (ANOVA), *p* ≤ 0.05, whereas ethidium bromide and cell membrane permeability data were analyzed using Graphpad prim 9, Student’s *t*-test *p* ≤ 0.05. Differential Gene Expression Analysis was conducted using DESeq software (v 3.14). A false detection rate (FDR)  ≤ 0.05 and the absolute value of Log_2_ ratio  ≥ 1 were used as the default threshold to identify significant DEGs.

## 4. Results and Discussion

Identification of *A. baumannii* mutants with impaired serum-associated antibiotic resistance.

Clinically defined antibiotic-susceptible *A. baumannii* strains have the remarkable ability to induce an expansive repertoire of drug efflux pumps during growth under physiological salt conditions, e.g., in pulmonary surfactant or human serum, which mediate resistance to physiologically achievable antibiotic concentrations within a patient [3,4,5,6,7]. More directly, and as previously shown, measures of cellular survival of the antibiotic susceptible strain 98-37-09 after incubation in conventional Luria-Bertani (LB) supplemented with various concentrations of minocycline, a front-line *A. baumannii* treatment option, revealed the strain to be susceptible to >1 µg mL^−1^ minocycline (Figure 1A) [35,36,37,38,39,40,41,42,43]. Conversely, the strain exhibited resistance to 2–4 µg mL^−1^ of the antibiotic during growth in human serum, correlating to or exceeding patient minocycline peak serum levels (~2.18 ± 0.44 µg mL^−1^ after 100 mg b.i.d. for 7–10 days) [44]. Moreover, the resistance phenotype was lost during growth in serum supplemented with the efflux pump inhibitor phenylalanyl arginyl β-naphthylamide (PaβN), indicating that serum-associated minocycline resistance is efflux-dependent (Figure 1A). This phenomenon, termed adaptive efflux mediated resistance (AEMR), has also been previously shown to modulate *A. baumannii* resistance to fluoroquinolones, aminoglycosides, and the glycylcline antibiotic, tigecycline, and is thought to account for one mechanism by which laboratory-defined antibiotic-susceptible strains resist antibiotic treatment within a patient [24].

To identify genetic determinants that modulate AEMR, a strain 98-37-09 transposon mutant library was screened for members that exhibited a loss of minocycline AEMR phenotype during growth in serum. To do so, individual members of the library were inoculated in human serum supplemented with 1 μg mL^−1^ minocycline (1× MIC in LB; 0.5× MIC in serum), and growth was monitored. We predicted that library members with proficient serum-inducible AEMR response would grow in the presence of the antibiotic, whereas mutants with a deficient AEMR response would not. Among the 6000 mutants evaluated, 5995 library members exhibited minocycline resistance during serum growth, suggesting that their AEMR process(es) were intact. Conversely, 5 transposon library members (0.1%) exhibited a loss of growth phenotype when cultured serum in the presence of minocycline, but normal growth in serum lacking minocycline. Indeed, as shown in Figure 1B, dose escalation studies measuring the viability of strain 98-37-09 and each of the five mutants in human serum supplemented with 0, 0.25, 0.5, 1.0, 2.0, or 4.0 µg mL^−1^ minocycline revealed that while wild type 98-37-09 cells resisted the antimicrobial properties of the antibiotic, each of the mutants exhibited a loss of resistance phenotype at minocycline concentrations ≥ 1–2 µg mL^−1^, indicating their AEMR response was impaired. Further, dose escalation testing in conventional LB media (Figure 1C) revealed that the susceptibility profile of each mutant of interest was virtually identical to that of wild-type 98-37-09 cells suggesting that the apparent loss of AEMR phenotype of each mutant in serum is not likely to be merely due to mutation of a gene that non-specifically leads to minocycline hypersusceptibility. Taken together, we considered that a subset of these mutants may harbor a transposon insertion within a genetic determinant that contributes to the *A. baumannii* AEMR response. Inverse PCR determined the transposon insertion site for each of the five mutants to be located within: a putative RND efflux pump protein (A1S_0008), a putative YhaK pirin-like protein (A1S_3277), isopropylmalate dehydratase (isomerase) subunit LeuD (A1S_0417), a putative ATP binding protein (A1S_0344), and a hypothetical protein (A1S_3826). These genes include three metabolic activity-associated genes, one efflux pump, and one pirin-like protein. The YhaK pirin-like protein possessed the strongest potential for a greater understanding of AEMR and thus was studied in more detail.

### 4.1. YhaK Impacts A. baumannii Serum-Inducible Efflux Properties

We were particularly intrigued that mutation of a putative YhaK pirin-like protein (A1S_3277) may be associated with an impaired AEMR response and chose to focus attention on this locus because pirin-like proteins have been associated with regulatory functions [45,46]. Thus, we considered that YhaK may play a key regulatory role in *A. baumannii* AEMR. To further evaluate whether YhaK contributes to AEMR, complementation was used to assess whether mutation of the pirin-like protein, as opposed to an unappreciated mutation within the strain, was responsible for the mutant’s apparent loss of AEMR phenotype. Accordingly, the *A. baumannii* YhaK putative open reading frame and corresponding predicted promoter region from strain 98-37-09 were cloned into plasmid pWH1266 to create pYhaK and transformed into YhaK-mutant cells. Because the plasmid backbone contains a tetracycline resistance determinant that would complicate minocycline resistance analyses, we switched to evaluating the AEMR phenotype of the wild-type, YhaK-mutant, and complementation strain using another antibiotic, levofloxacin, which has been previously been shown to be affected by AEMR [26].

Consistent with previous results, testing revealed that wild-type 98-37-09 cells were susceptible to ≥0.125 µg mL^−1^ levofloxacin during growth in LB media, but resistant to a ≥4-fold increase in the antibiotic (0.5 µg mL^−1^) during AEMR-inducing growth in human serum (Figure 2A) [26]. Conversely, the YhaK-mutant exhibited a loss of resistance phenotype, displaying susceptibility to 0.5 μg mL^−1^ levofloxacin during serum growth. The mutant’s apparent loss of levofloxacin resistance during growth in human serum was unaffected by the presence of the shuttle vector pWH1266 but could be rescued by plasmid pYhaK harboring a wild-type copy of the gene, suggesting that the YhaK pirin-like protein contributes to *A. baumannii* AEMR (Figure 2A). Attempts to create a clean strain 98-37-09 *yhaK* deletion were unsuccessful. Thus, to verify the effects of YhaK on AEMR in an independent strain background, we obtained *A. baumannii* strains AB5075-UW (wild type) and an isogenic YhaK-mutant, AB5075-YhaK, from the Manoil transposon mutant library [30] and assessed their levofloxacin susceptibility profiles during AEMR-inducing serum growth conditions. Results revealed that while the wild-type strain, AB5075-UW was intrinsically more resistant to levofloxacin than strain 98-37-09, and it was susceptible to ≥16 µg mL^−1^ levofloxacin during growth in LB media and exhibited a ≥4-fold increase in resistance to (64 µg mL^−1^) during growth in human serum suggesting it undergoes AEMR (Figure 2B). Conversely, the YhaK-mutant strain exhibited loss of resistance to 64 µg mL^−1^ levofloxacin during serum growth, which could be restored by complementation with pYhaK harboring a wild-type copy of the *yhaK* gene but not by the plasmid vector alone. Taken together, these results indicated that the YhaK pirin-like protein is required for levofloxacin resistance during serum growth in at least two strain backgrounds, presumably by participating in the AEMR response.

To determine whether the apparent loss of AEMR phenotype of YhaK-mutant cells does indeed correspond with a loss of the strain’s efflux properties, standard ethidium bromide (EtBr) efflux assays were used to compare the efflux capacity of wild-type 98-37-09 and YhaK-mutant cells during growth in LB (non-AEMR) and human serum (AEMR inducing) conditions. The premise for such EtBr assays is that efflux-deficient cells will accumulate ethidium bromide and display high corresponding fluorescence in comparison to efflux-proficient cells that extrude the dye [47]. As previously reported, EtBr assays of wild-type cells grown in LB and serum (Figure 3A), revealed the strain exhibited increased EtBr accumulation during growth in LB media (3332.14 ± 38.79 Relative Fluorescent units; RFUs) in comparison to growth in human serum (2576.17 ± 40.88 RFU), confirming earlier studies that showed serum-induced AEMR antibiotic resistance correlates with increased efflux capacity [26]. Conversely, studies of the YhaK-mutant indicated that it exhibits higher ethidium bromide accumulation (2959.5 ± 51.47 RFU) during serum growth in comparison to wild-type cells, suggesting that the apparent loss of AEMR due to *yhaK*-mutation correlates with a loss of serum-inducible efflux capacity (Figure 3A). To distinguish whether the observed differences in EtBr accumulation between serum-grown wild-type and mutant cells may be attributable to alterations in cell membrane permeability, as opposed to differences in the efflux properties of the two strains, standard 1-*N*-phenylnapthylamine (NPN) cell permeability assays were performed on serum-grown wild-type and mutant cells. As shown in Figure 3B, no significant difference in NPN cell permeability was observed between the two strains during serum growth; the detergent triton X-100 was used as a positive control to ensure the assay was indeed capable of measuring increased cell permeability of both strains in the presence of human serum. Taken together, these results indicate that the apparent loss of AEMR associated with the mutation of *yhaK* is likely to be due to limited efflux potential as opposed to compromised membrane permeability.

### 4.2. YhaK Impacts A. baumannii Serum-Inducible Adaptive Efflux-Mediated Antibiotic Resistance

While ethidium bromide has historically served as an excellent surrogate for measuring the overall efflux properties of bacterial cells, we set out to more conclusively assess whether YhaK mutation affects *A. baumannii* antibiotic extrusion. Accordingly, a fluorescent isomer of levofloxacin, ofloxacin, was created (DMACA-Oflox; Appendix A) as a tool to more directly measure AEMR-mediated antibiotic accumulation within wild-type and YhaK-mutant cells. We chose DMACA because it is a low molecular weight fluorescent molecule that has successfully been shown to measure bacterial quinolone internalization and efflux, while also retaining antibiotic activity [48]. As a prerequisite for doing so, we assessed whether wild-type cells exhibited AEMR-associated resistance to ofloxacin and/or the fluorescent derivative.

As shown in Figure 4A, wild-type 98-37-09 cells were susceptible to ≥0.25 µg mL^−1^ ofloxacin during growth in LB but resistant to ≥1 µg mL^−1^ of the antibiotic during growth in serum, suggesting that ofloxacin is impacted by AEMR. Conversely, such resistance was ablated for the mutant during growth in human serum. More directly, the YhaK-mutant strain was found to be susceptible to ≥0.25 µg mL^−1^ ofloxacin during growth in LB but showed a loss of resistance to 1 µg mL^−1^ of the antibiotic during serum growth, indicating that YhaK partially impacts serum growth associated ofloxacin resistance (Figure 4A). Similarly, while fluorescence-labeling seemed to reduce ofloxacin’s antimicrobial potency, wild-type cells displayed resistance to ≥200 µg mL^−1^ DMACA-Oflox during serum growth whereas serum-grown YhaK-mutant cells exhibited a loss of resistance to 200 µg mL^−1^ DMACA-Oflox (Figure 4B). Moreover, the YhaK-mutant phenocopied DMACA-Oflox-treated serum-grown wild-type cells in the presence of the known *A. baumannii* AEMR inhibitor, AE0003 (structure provided in Appendix A), indicating that mutation of the strain’s YhaK protein renders them AEMR-deficient (Figure 4C) [33].

### 4.3. YhaK-Mutant Cells Accumulate DMACA-Oflox

Confocal microscopy was used to determine whether the apparent loss of DMACA-Oflox adaptive efflux-mediated resistance in YhaK-mutant cells did indeed correlate with increased cellular accumulation of the ofloxacin derivative in comparison to AEMR-proficient wild-type cells. As expected, confocal microscopy revealed that wild-type cells accumulate DMACA-Oflox during growth in LB media (448 ± 50 relative fluorescent units; RFU) but display a 15.4-fold reduction in accumulation (29 ± 4 RFU) during AEMR-inducing serum growth conditions (Figure 5A,B). Tests of the YhaK mutant revealed (Figure 5C,D) that the strain accumulates the antibiotic-derivative to wild-type levels during growth in LB (484 ± 65 RFU) yet only results in a 5.7-fold reduction in accumulation (84 ± 16 RFU) during growth in human serum, suggesting that the mutant’s loss of DMACA-Oflox AEMR correlates with increased intracellular antibiotic accumulation. Figure 5E quantifies the average confocal microscopy fluorescence signals for all assay conditions. Moreover, as shown in Figure 6, the reduced DMACA-Oflox accumulation of serum-grown wild-type cells could, at least in part, could be reversed by the *A. baumannii* efflux pump inhibitor AE0003, indicating that DMACA-Oflox allows measures of the organism’s adaptive efflux-mediated resistance.

Similarly, flow cytometry studies indicated that YhaK-mutant cells exhibited increased DMACA-Oflox accumulation during growth in human serum in comparison to wild-type cells. More directly, wild-type and YhaK-mutant cells grown in LB-supplemented 75 µM DMACA-Oflox produced side scatter distributions of ≥10^4^ RFU indicative of high accumulation of the antimicrobial, plotted as light blue and light green within Figure 7A,B, respectively. However, during AEMR-inducing serum growth, wild-type cells exhibited a significant reduction in DMACA-Oflox association, resulting in a cell population shift with approximately a 10-fold decrease in RFU (comparison of light blue to dark blue) resulting in a phenotype that could partially be rescued by cellular treatment with the AEMR inhibitor AE0003 (Figure 7C,D), indicating flow cytometry allows measures of the organism’s AEMR phenotype. In direct agreement with confocal microscopy, tests of the serum-grown YhaK-mutant cells revealed an intermediate cell population shift of approximately 5-fold reduction in RFU units (comparison of light green to dark green), suggesting that the mutant has increased DMACA-Oflox association in comparison to wild-type cells during growth in human serum (Figure 7A,B).

### 4.4. RNA Sequencing of Wild-Type and YhaK-Mutant Cells during Growth in Human Serum

Recognizing that pirin-like proteins display regulatory effects and hence YhaK’s putative contributions to AEMR may be regulatory in nature, we set out to assess the protein’s cellular effects using RNA sequencing. As a first step toward doing so, we assessed the protein’s expression within wild-type cells during growth in LB and human serum, both in the absence and presence of levofloxacin using quantitative RT-PCR. Results (Appendix A) revealed that the YhaK gene is upregulated approximately 2-fold in cells grown in human serum in comparison to growth in LB media. Further, the presence of the antibiotic stimulated gene transcription approximately 4-fold, suggesting that its regulatory effects may be optimally measured by comparing the transcriptomes of wild-type and YhaK-mutant cells during growth in human serum supplemented with levofloxacin.

Comparison of the gene expression properties of wild-type and YhaK-mutant cells during these conditions revealed that 333 genes were upregulated and 320 genes were down-regulated at least 2-fold (FDR ≤ 0.05) in wild-type cells in comparison to the mutant, suggesting that mutation of the YhaK protein results in pleiotropic effects. To focus attention on defining the potential mechanism(s) by which YhaK may modulate adaptive efflux-mediated antibiotic resistance, Figure 8 provides an overview of the predominant KEGG gene ontology groups differentially expressed in a YhaK-dependent manner. Within YhaK expressing wild-type cells, genes involved in transcription, cell signaling, and post-translational modification were highly expressed in comparison to mutant cells, as were systems belonging to numerous transport modalities including nucleotides, inorganic ions, co-enzymes, and amino acids. Among the most repressed within wild-type cells where genes are associated with translation machinery, secondary metabolites, lipid transport, and energy production.

Further assessment of individual components of these ontology groups provided several plausible mechanisms by which YhaK may modulate adaptive efflux-mediated resistance. In perhaps the most straightforward scenario, one could imagine that YhaK stimulates the expression of antibiotic drug efflux pumps during growth in human serum, thereby directly accounting for an AEMR phenotype. While the majority of previously characterized *A. baumannii* antibiotic efflux pumps were not differentially expressed between the two strains, 15 transport pumps, 11 of which are predicted, have been associated with antibiotic resistance within other organisms, where they were found to be upregulated at least 2-fold in wild-type cells during serum growth in the presence of levofloxacin in comparison to YhaK-mutant cells (Figure 9). These include a putative MATE drug efflux pump (3.5×; locus A1S_3420), four MFS pump proteins (2.1–8.2×; A1S_0801, A1S_0802, A1S_2326 and A1S_2327), four ATP-binding cassette pump proteins (3–4.2×; loci A1S_1611, A1S_1613, A1S_1786, and A1S_2299), and ammonium transporters (4.8–5.2×; loci A1S_021 and A1S_2748; ref. [49]). In a second possibility, YhaK may be responsible for activating cellular efflux capability post-translationally. Indeed, studies have revealed that drug efflux pumps can be post-translationally regulated and/or require modification to become activated, and as shown in Figure 8, post-translational regulatory processes are among the most highly upregulated within wild-type as opposed to YhaK-mutant cells during serum growth in the presence of levofloxacin [50]. Third, efflux pumps require energy for activity, and two ammonium transporters (above) were found to be expressed in a YhaK-dependent manner. A fourth possibility is that YhaK’s affects serum-associated antibiotic resistance in a mechanism that does not directly involve modulating cellular efflux. To that end, results revealed that ClpP protease is upregulated 2.3-fold within wild-type, as opposed to YhaK-mutant cells. ClpPX has been shown to affect antibiotic tolerance during stress conditions in other Gram-negative pathogens, such as *E. coli* [51].

Regardless of the molecular mechanism(s), taken together, these results suggest that YhaK is likely to contribute to *A. baumannii*’s AEMR phenotype. *A. baumannii* YhaK (accession # COG1741) is predicted to code for a 290-amino acid gene product that is annotated as pirin-like protein within the cupin superfamily based on primary sequence alignment. BLAST search indicates that the protein is highly conserved across 176 publicly available *A. baumannii* genomes (≥97% amino acid identity) and shares 55–70% amino acid identity with proteins of other Gram-negative pathogens including *P. aeruginosa*, *K. pneumonia*, *Yersinia pestis*, and *Salmonella enterica*, suggesting that its cellular functions may be conserved across species. Until recently, pirin-like protein function(s) have been poorly understood, with reports spanning both enzymatic and non-enzymatic activities. Arguably, most information about the protein family derives from studies of human pirin (hPirin) and the *E. coli* pirin protein, YhhW. hPirin has been shown to bind transcription factors NF-I and Bcl-3 in vivo, suggesting that pirins act as transcription co-factors to regulate gene transcription [52,53]. In the case of the latter, hPirin is thought to stabilize quaternary complexes between Bcl-3, NF-ĸB, and promoter sequences [54]. To our knowledge, no studies have evaluated whether *E. coli* YhhW displays regulatory effects, yet the tertiary structures of both human pirin and YhhW have been determined and are similar to quercetin 2,3-dioxygenase, which led to the finding that both hPirin and YhhW display quercetinase activity [55]. It is not clear whether the hPirin’s enzymatic activity is associated, either directly or indirectly, with the protein’s regulatory function(s). More recently, it was found that although the *E. coli* YhaK pirin-like protein shares overall topology with hPirin and YhhW, it does not seem to exhibit quercetinase activity, presumably due to the absence of three conserved pirin-like protein amino acid residues (His57, His59, and His101) that are thought to govern enzymatic function [56]. Hence, it has been proposed that the *E. coli* protein should be reannotated solely as a bicupin to distinguish it from the *E. coli* YhhW pirin-like protein [56].

Alignments of the *A. baumannii* YhaK protein that is the focus of these studies indicates that the protein best fits and shares the conserved residues of hPirin and YhhW pirin family hypothesized to mediate quercetinase activity, suggesting that the protein may display both regulatory and enzymatic activities that have been associated with other pirins (Appendix A). To that end, RNA sequencing of wild-type and YhaK-mutant cells during AEMR-inducing growth conditions revealed that *A. baumannii* YhaK is required for wild-type levels of expression of multiple transporter systems including those that are ostensibly associated with antibiotic efflux, such as a multidrug and toxic compound extrusion (MATE) family protein and several ATP-binding cassette (ABC) pumps. It remains to be seen whether the protein’s regulatory effects are direct or indirect. Interestingly, YhaK did not seem to affect the expression of the RND-pump protein identified in this study (A1S_0008) or pumps that have been previously characterized to mediate antibiotic efflux. It will be interesting to learn whether point mutations within YhaK residues that are predicted to be required for quercetinase activity affect the protein’s regulatory effects.

While it is intriguing to consider that *A. baumannii* YhaK’s contribution to AEMR is via transcriptional regulation of efflux pumps, other regulatory effects were also noted, as described above, suggesting that mutation of the protein leads to pleiotropic effects, and hence YhaK’s role in modulating AEMR may be indirect. Regardless of the mechanism, the results presented indicate that the protein is a determinant of the organism’s serum-inducible AEMR response to at least two classes of antibiotics, the fluoroquinolones and tetracyclines. As such, one can imagine that small molecule inhibitors of YhaK activity may have therapeutic utility as adjuvants, which improve the therapeutic potential of these antibiotics toward *A. baumannii*. In that regard, it will be important to evaluate whether YhaK is required for AEMR toward other classes of antibiotics and whether these effects are conserved across contemporary *A. baumannii* clinical isolates in future studies.

The transposon mutant library screen used to identify YhaK also revealed that mutation of four other loci, including a putative RND efflux pump protein, isopropylmalate dehydratase, a putative ATP binding protein, and a hypothetical protein, may also affect *A. baumannii*’s AEMR response. Ostensibly, the RND efflux pump system may play a direct role by extruding antibiotics during AEMR-inducing conditions. While the putative contributions of the other gene products are more obscure, we expect that as components of the system are identified, verified, and characterized, their roles will become clearer. Admittedly, a limitation of the transposon screening approach used is that it, by definition, excludes evaluating the contributions of essential gene products. In that regard, YhaK may serve as a biochemical and/or genetic tool to identify additional components of *A. baumannii* AEMR.

## Figures and Tables

**Figure 1 antibiotics-12-01173-f001:**
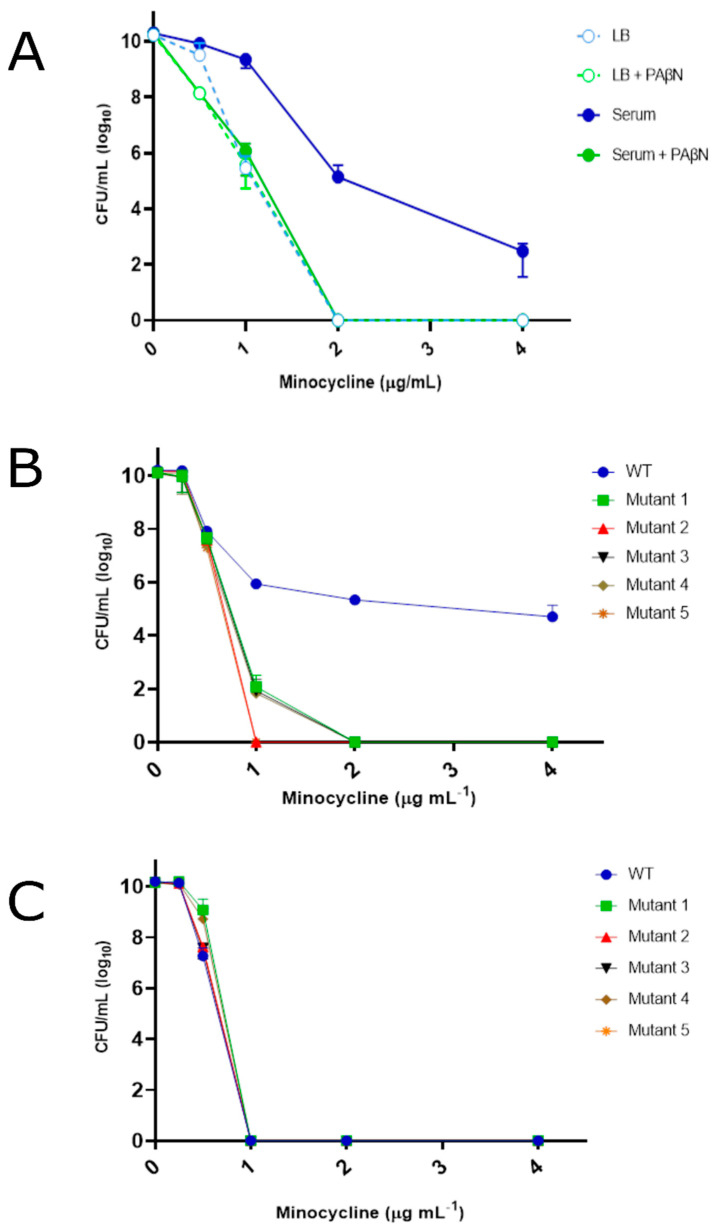
Growth phenotype of *A. baumannii* strains in Luria Bertani and human serum supplemented with minocycline. Plotted are the colony forming units (CFUs) of *A. baumannii* strain 98-37-09 (WT) during growth in LB media or human serum supplemented with 0, 0.25, 0.5, 1, 2, or 4 µg mL^−1^ minocycline +/− the efflux pump inhibitor PaβN (Panel **A**). Putative AEMR mutants following incubation in human serum (Panel **B**) or LB media (Panel **C**) supplemented with 0, 0.25, 0.5, 1, 2, or 4 µg mL^−1^ minocycline; standard deviations shown.

**Figure 2 antibiotics-12-01173-f002:**
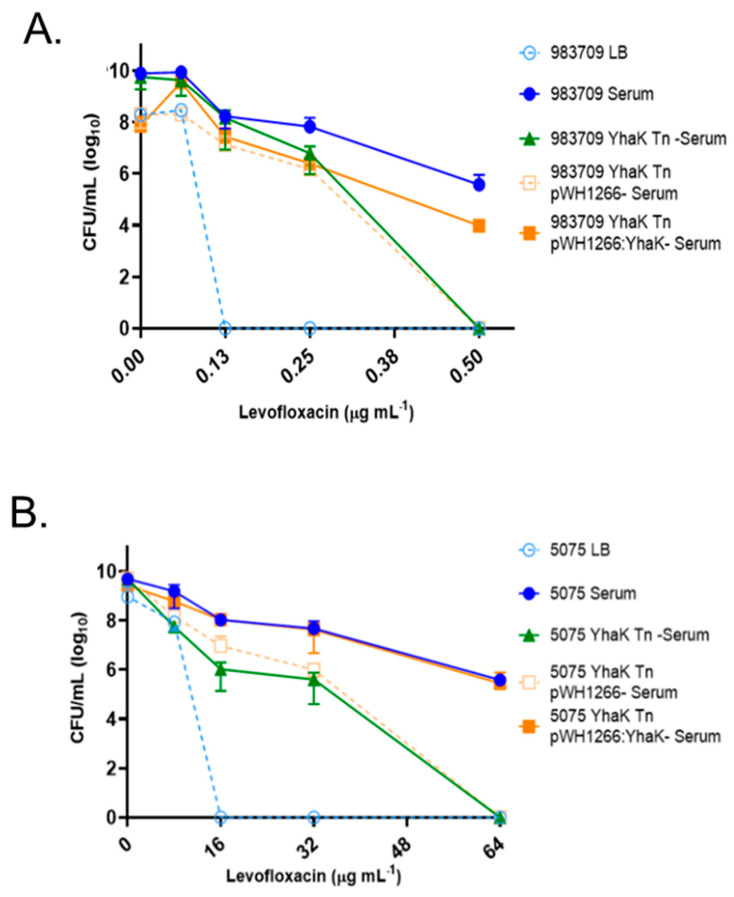
Growth phenotype of *A. baumannii* strains in Luria Bertani and human serum supplemented with levofloxacin. (**A**) Plotted are the colony-forming units (CFUs) of *A. baumannii* strain 98-37-09 (WT), YhaK transposon mutant (YhaK Tn), and YhaK transposon mutant containing plasmid pWH1266 or pWH1266::YhaK following incubation in human serum supplemented with 0, 0.06, 0.125, 0.25, or 0.5 µg mL^−1^ levofloxacin; standard deviations shown. (**B**) Plotted are the colony-forming units (CFUs) of *A. baumannii* strain AB5075-UW (WT), YhaK transposon mutant (YhaK Tn), and YhaK transposon mutant containing plasmid pWH1266 or pWH1266::YhaK following incubation in human serum supplemented with 0, 8, 16, 32, or 64 µg mL^−1^ levofloxacin; standard deviations shown.

**Figure 3 antibiotics-12-01173-f003:**
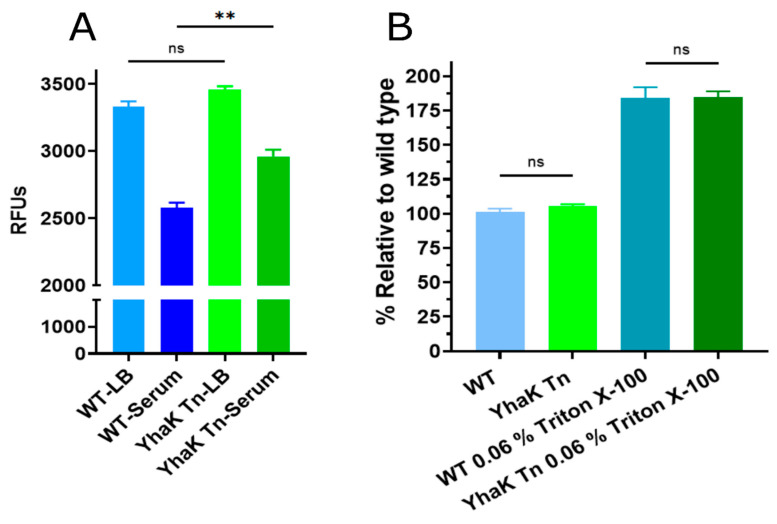
Ethidium bromide accumulation and 1-*N*-phenylnapthylamine cell permeability assays. (Panel **A**) Plotted are the relative fluorescent units (RFUs) of ethidium bromide-treated *A. baumannii* strains 98-37-09 (WT) and isogenic YhaK-mutant (YhaK Tn) following growth in LB or human serum. (Panel **B**). Plotted are the RFUs of WT and YhaK Tn cells following growth in human serum and treatment with the cell permeability dye 1-*N*-phenylnapthylamine in the absence or presence of 0.06% Triton X-100 detergent; standard deviations and significance shown (Student’s *t*-test, ** *p* ≤ 0.01; not significant (ns)).

**Figure 4 antibiotics-12-01173-f004:**
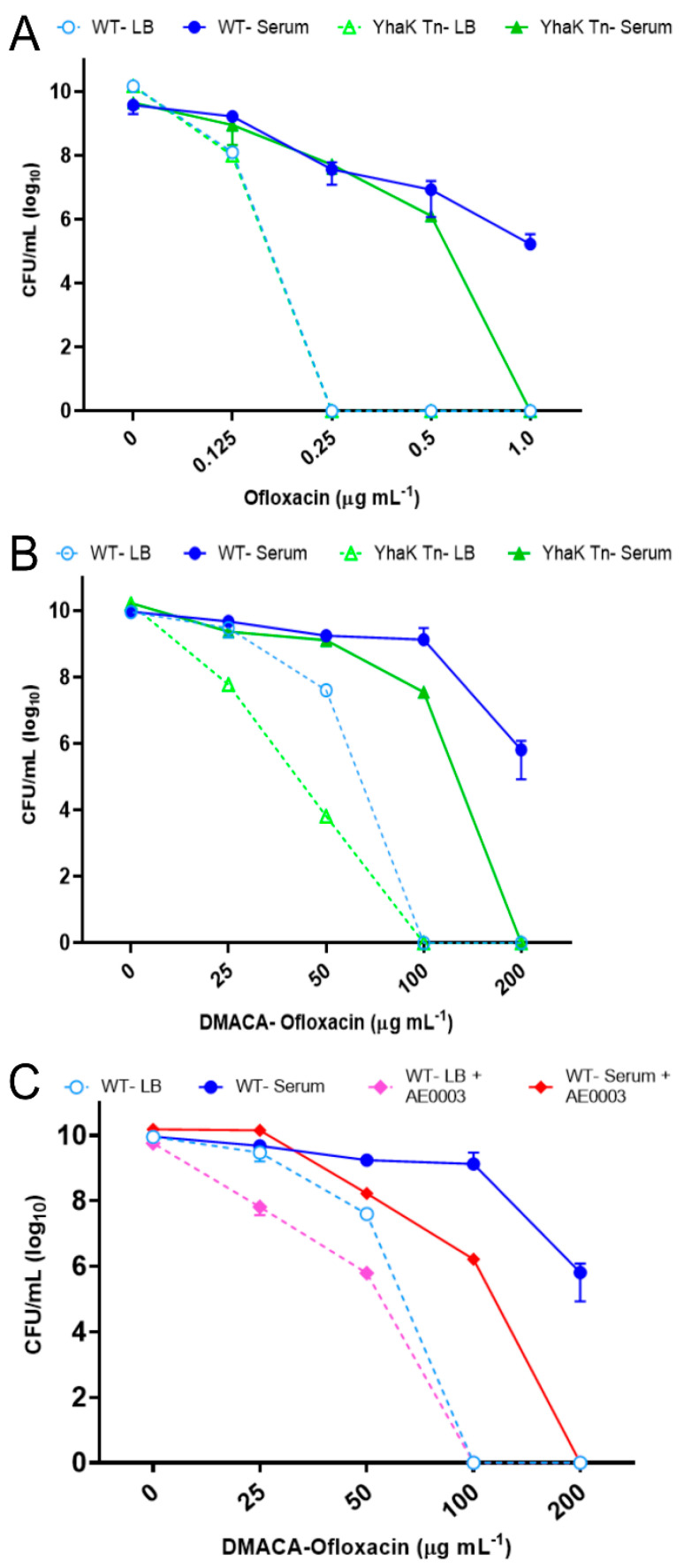
Growth phenotype of *A. baumannii* strains in Luria Bertani and human serum supplemented with ofloxacin or DMACA-ofloxacin. Plotted are the colony-forming units (CFUs) of *A. baumannii* strain 98-37-09 (WT) and YhaK Tn mutant (Yhak Tn) following incubation in LB media or human serum supplemented with the indicated concentration of the antibiotic olfoxacin (Panel **A**) or DMAC-olfoxacin (Panel **B**). (Panel **C**) Plotted are CFUs of WT cells following incubation in LB or human serum treated with DMACA-ofloxacin in the absence or presence of the efflux pump inhibitor AE0003; standard deviations are shown.

**Figure 5 antibiotics-12-01173-f005:**
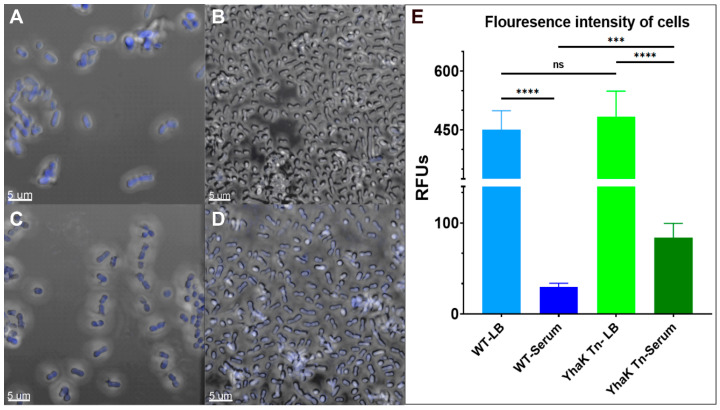
Confocal microscopy of DMACA-Ofloxacin treated wild-type and YhaK Tn-mutant cells following growth in LB or human serum. Shown are stacked confocal images of DMACA-Oflox-treated WT cells following growth in LB (Panel **A**) or human serum (Panel **B**) or YhaK-mutant cells following growth in LB (Panel **C**) or human serum (Panel **D**). (Panel **E**). Averaged fluorescence intensity (*n* = 20) graphed; standard deviation shown; statistical difference (One-way ANOVA, *** *p* ≤ 0.001, **** *p* ≤ 0.0001; not significant (ns)).

**Figure 6 antibiotics-12-01173-f006:**
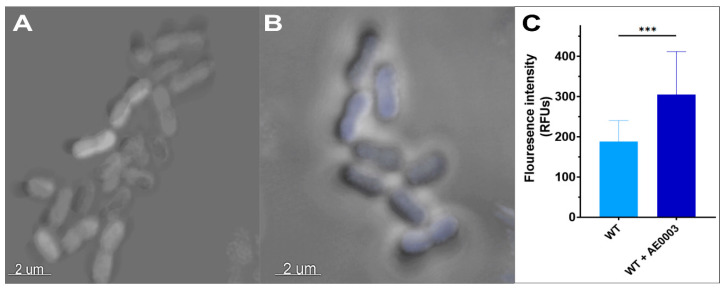
Confocal microscopy of DMACA-Ofloxacin treated wild-type cells in the presence of the efflux pump inhibitor AE0003. Stacked confocal images of DMACA-Oflox treated serum grown WT cells in the absence (Panel **A**) or presence (Panel **B**) of the efflux pump inhibitor AE0003. (Panel **C**). Averaged fluorescent signal of untreated and AE0003-treated serum-grown WT cells; standard deviation shown (*n* = 20); *** indicates statistical difference (Student’s *t*-test, *p* ≤ 0.001).

**Figure 7 antibiotics-12-01173-f007:**
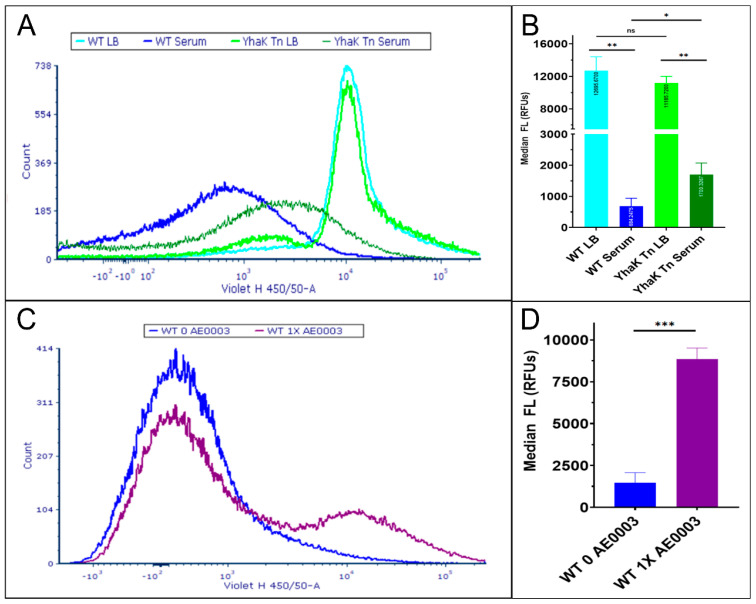
Flow cytometry of DMACA-Ofloxacin-treated cells. (Panel **A**) Plotted are the intensity measures of DMACA-Ofloxacin-treated WT cells during growth in LB (light blue) or human serum (dark blue) or YhaK-mutant (Yhak-Tn) cells following LB (light green) or serum (dark green) growth, with averaged median fluorescence from duplicate studies shown in (Panel **B**). (Panel **C**) Plotted are the fluorescence intensity and side scatter of DMACA-Oflox treated serum grown WT cells in the absence (dark blue) or presence (purple) of the *A. baumannii* efflux pump inhibitor AE0003, with averaged median fluorescence from duplicate studies shown in (Panel **D**). Standard deviation shown; statistical significance (One-way ANOVA, * *p* ≤ 0.05, ** *p* ≤ 0.01, *** *p* ≤ 0.001 and not significant (ns)).

**Figure 8 antibiotics-12-01173-f008:**
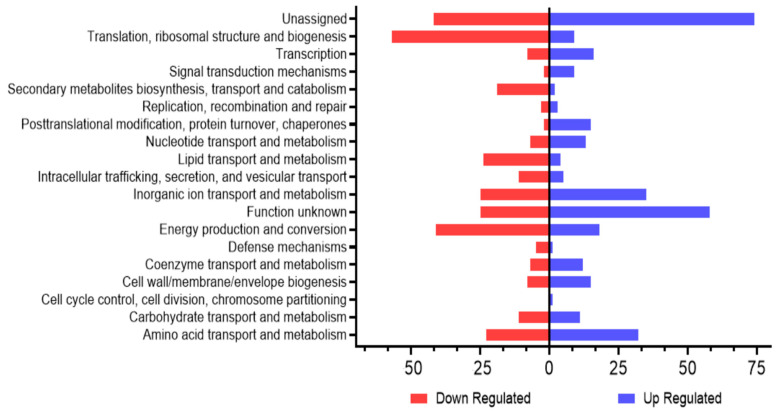
*A. baumannii* WT and YhaK Tn Transcriptome analysis: KEGG gene ontology groups. A number of genes displaying > 2-fold alteration in wild-type compared to YhaK-mutant cells by gene ontology.

**Figure 9 antibiotics-12-01173-f009:**
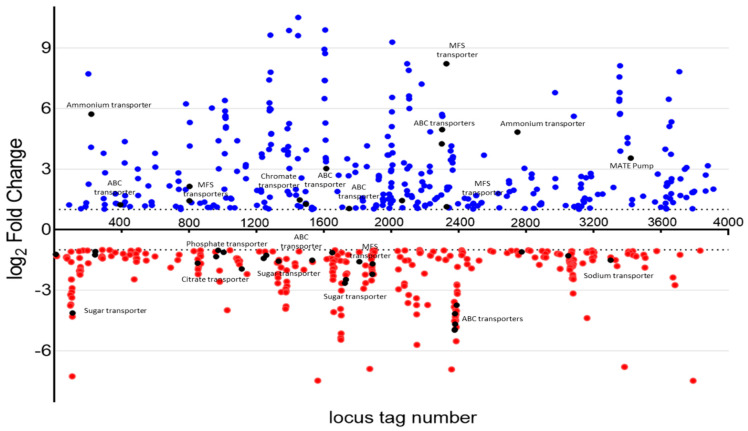
*A. baumannii* WT and YhaK Tn Transcriptome analysis: differentially expressed genes. Scatter plot of differentially expressed genes within wild-type and YhaK-mutant cells during grown in serum supplemented with levofloxacin; log_2_ fold change in expression for each gene was plotted against gene locus tag number. Loci determined to be upregulated (greater transcript titers) in wild-type cells in comparison to YhaK-mutant cells are labeled in blue, whereas downregulated genes are labeled red. Loci annotated to code for known or putative transporters are labeled in black; known transporters are also labeled.

**Table 1 antibiotics-12-01173-t001:** Strains and plasmids used in this study.

Strains and Plasmids	Relevant Genotype/Phenotype	Source
* A. baumannnii *		
98-37-09	Wild type	[28]
98-37-09 YhaK	98-37-09 *yhaK*::EZTn5	[28]
98-37-09 YhaK pWH1266	98-37-09 *yhaK*::EZTn5; pWH1266	This study
98-37-09 YhaK pYhaK	98-37-09 *yhaK*::EZTn5; pYhaK	This study
AB5075-UW	Wild type	[30]
AB5075 YhaK	ABUW_0207-119::T26	[30]
AB5075 YhaK pWH1266	ABUW_0207-119::T26; pWH1266	This study
AB5075 YhaK pYhaK	ABUW_0207-119::T26; pYhaK	This study
* E. coli *		
OneShot^®^ TOP10	F-*mcrA* Δ(*mrr-hsd*RMS-*mcr*BC) Φ80*lac*ZΔM15 Δ*lac*X74 *rec*A1 *ara*D139 Δ(*araleu*)7697 *gal*U *gal*K *rps*L (StrR) *end*A1 *nup*G	Thermo Fisher
Plasmids		
pWH1266	pBR322 derivative; tet^R^ and amp^R^	[29]
pYhaK	pWH1266 containing 98-37-09 YhaK	This study

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
