# Peer review of "Genetic Determinants of Acinetobacter baumannii Serum-Associated Adaptive Efflux-Mediated Antibiotic Resistance"

_antibiotics, 2023, doi:10.3390/antibiotics12071173_

Round 1

Reviewer 1 Report

The current work examines the genetic determinants and causation of antibiotic resistance within the nosocomial pathogen A. baumannii. The authors offer evidence that antibiotic resistance in A. baumannii is due to serum-associated Adaptive Efflux Mediated Resistance (AEMR), and that the AEMR phenotype manifests in part due to the yhaK gene which encodes a pirin-like protein. The results show that strains in which the YhaK locus was disrupted exhibit reduced serum-associated resistance to several antibiotics (minocycline, levofloxacin, and ofloxacin), as well as reduced efflux potential in ethidium bromide efflux assays. The strains also exhibit increased intracellular accumulation of fluorescent-tagged levofloxacin in microscopy and flow-cell assays. Mutation of the YhaK protein renders the strains to be AEMR deficient, thus serving as a strong indicator of correlation between YhaK protein function, AEMR, and antibiotic resistance. 

The background information provided by the authors sufficiently covers the pre-existing literature and builds upon the current knowledge of serum-associated AEMR in A. baumannii. The authors point out the importance of the work by noting that A. baumannii is designated as one of the 6 bacterial pathogens of greatest concern according to ESKAPE. The study design is very good. The methods and materials are highly descriptive and refer to methods in the previous literature as appropriate. I have no further recommendations for improvement. 

Author Response

Reviewer 1: 

The reviewer stated: “The background information provided by the authors sufficiently covers the pre-existing literature and builds upon the current knowledge of serum-associated AEMR in A. baumannii. The authors point out the importance of the work by noting that A. baumannii is designated as one of the 6 bacterial pathogens of greatest concern according to ESKAPE. The study design is very good. The methods and materials are highly descriptive and refer to methods in the previous literature as appropriate. I have no further recommendations for improvement.”

Response:  The authors are grateful for the reviewer’s comments. We believe the work lays the foundation for future studies into the development of therapeutics that target A. baumannii’s AEMR.

Reviewer 2 Report

1. Brief summary

I value the opportunity to review this interesting manuscript. The aim of the paper entitled “Genetic determinants of Acinetobacter baumanniiserum-associated adaptive efflux mediated antibiotic resistance” may appropriate for consideration of publication in the Antibiotic. However, there are major concern about this manuscript.

2. General concept comments

- The data reported in the manuscript is scientifically interesting. Unfortunately, just 32.84% (22/67 publications) of current citations in the preceding five years, including 2018 (2 article), 2019 (9 article), 2020 (5 articles), 2021 (5 articles) and 2022 (1 articles), are cited. Please consider updating the references.

- The introduction section should be improved. Since, all of the paragraphs are too general and lengthy. The research objectives also cannot follow from the introduction.

- There is no a geometric progression of the concentration in growth phenotype of A. baumannii strains in Luria Bertani and human serum supplemented with levofloxacin (Panel A). Why did you choose these concentration (0, 0.13, 0.25, 0.38 and 0.50 ug/mL) ?

- More specifics are required in your section on materials and methods. Please describe the statistics that were used in this study. Statistical evaluation is a valuable tool in experimental research and is required for proper interpretation of data collected through experiments.

- Only the results of your study should be included in the results section. As a general rule, any information that does not convey the study's direct findings or outcome should be omitted from this part. In addition, the citations may not be required in the experiment results.

- More specifics are required in your discussion. The writers identify the relevance of the results and critically assess all of your data in this section. These interpretations are frequently derived by comparing and contrasting the findings of previous investigations, hence references to the studies addressed.

- I would suggest the author to describe the study's limitations in the discussion section.

- I would like to propose adding a concluding section. It is important to summarize in a few paragraphs the main revelation from the significant evidence that was collected. Additionally, this research's conclusion section requires a suggestion for further research.

- Please arrange the keywords alphabetically for a standardized presentation.

- The full genus name must be spelled out the first time it is used with a new species. On subsequent mentions of a species, the genus should be abbreviated. All scientific bacterial names should be italicizedPlease double-check the name and symbol of each gene/protein also. These acronyms are not emphasized.

- To ensure clarity and consistency, use the full wording first, followed by the abbreviation to be used later in brackets. On following occasions, merely use the acronym.

- Normal font size should be used for writing the International System of Units.

Finally, I am grateful for the chance to review this submission. Although the topic is valuable and interesting, further interpretation and motivation is required. Wishing the authors success for their paper publication, I hope the suggestions are beneficial.

Author Response

Reviewer 2:

The reviewer stated: "The data reported in the manuscript is scientifically interesting. Unfortunately, just 32.84% (22/67 publications) of current citations in the preceding five years, including 2018 (2 article), 2019 (9 article), 2020 (5 articles), 2021 (5 articles) and 2022 (1 articles), are cited. Please consider updating the references.”

Response: Investigating the effect of human serum on antibiotic efflux activity and antibiotic resistance is a small niche and thus there are not many papers within the last 5 years which cover this topic.  We have directly referenced the appropriate background studies.

The reviewer stated: “The introduction section should be improved. Since, all of the paragraphs are too general and lengthy. The research objectives also cannot follow from the introduction.”

Response: This statement is somewhat confusing, as we specifically introduce the readers to the multitude of known A. baumnnii efflux pumps, the concept that additional pumps are likely to contribute to antibiotic resistance, and fact that such efflux pumps are turned on during growth in biologically relevant growth conditions (adaptive efflux mediated resistance; AEMR).  Our goal of this study is to define genetic determinants of AEMR.  While we agree with the reviewer regarding expanding the discussion section, we feel that this comment is a bit stylistic.

The reviewer stated: “There is no a geometric progression of the concentration in growth phenotype of A. baumannii strains in Luria Bertani and human serum supplemented with levofloxacin (Panel A). Why did you choose these concentration (0, 0.13, 0.25, 0.38 and 0.50 ug/mL) ?”

Response:  This is an astute question in reference to Figure 2A.  The MIC of A. baumannii strain 983709 in LB media was determined to be 0.125 ug/mL levofloxacin.  Thus, we evaluated bacterial growth in serum at 0, 0.5, 1, 2,and 4X the strain’s MIC in LB (0, 0.0625, 0.125, 0.25 and 0.5 ug/ml levofloxacin).  When plotting the data in the same format as other figures the x-axis Prism software automatically adds 0.38 to the x-axis to maintain spacing.  No data points were collected at 0.38 ug/ml levofloxacin.  We have clarified this in the Figure legend “in human serum supplemented with 0, 0.06, 0.125, 0.25, or 0.5 ug ml levofloxacin” (lines 474-475 of the revised text).

The reviewer stated:  “More specifics are required in your section on materials and methods. Please describe the statistics that were used in this study. Statistical evaluation is a valuable tool in experimental research and is required for proper interpretation of data collected through experiments.”

Response:  While statistical approaches are included in corresponding Figure legends we agree with the reviewer that they should also be included within the relevant Materials and Methods sections of the manuscript as well.  Accordingly, we have provided statistical tests used when analyzing various experimental results with the Material and Methods section (lines 220-221, 234-235, 314-315, 327-328, and 343-346,  of the revised text).

The reviewer stated: Only the results of your study should be included in the results section. As a general rule, any information that does not convey the study's direct findings or outcome should be omitted from this part. In addition, the citations may not be required in the experiment results.

Response:  We agree but are honestly confused by the reviewer’s comment.  We assume the reviewer is referring to the results section “RNA sequencing of wild type and YhaK mutant cells during growth in human serum” in which we feel that it is important to put into context why we highlighted subsets of the >600 genes identified to be differentially expressed between strains.   

The reviewer stated:  “More specifics are required in your discussion. The writers identify the relevance of the results and critically assess all of your data in this section. These interpretations are frequently derived by comparing and contrasting the findings of previous investigations, hence references to the studies addressed.” Further s/he stated “I would suggest the author to describe the study's limitations in the discussion section.”

Response:  Again, we are honestly confused by the reviewer’s comment.  This paper identifies a putative pirin-like-protein to be a driver of A. baumannii AEMR.  We go into great detail with regard what is known about pirin proteins in other species and discuss how the protein may be modulating AEMR as well as compare the amino acid residues of proteins to allow informing conclusions to be met. None the less, we have included text to indicate that a limitation of the study is that AEMR is known to affect at least 5 chemical classes of antibiotics, only two of which have been directly investigated here and we have not yet evaluated the impact of YhaK mutation in contemporary A. baumannii clinical isolates (lines 701-703 of revised text).  Additionally, we have added a paragraph describing a limitation of using transposon mutants is that it does not allow study of essential genes.  Further, we discuss the use of YhaK as a biochemical and/or genetic tool to identify such essential gene products (lines 707-717 of revised text).

The reviewer stated:  “Please arrange the keywords alphabetically for a standardized presentation.”

Response: We have now arranged the keywords alphabetically.

.

The reviewer stated:  “The full genus name must be spelled out the first time it is used with a new species. On subsequent mentions of a species, the genus should be abbreviated. All scientific bacterial names should be italicized. Please double-check the name and symbol of each gene/protein also. These acronyms are not emphasized.”

Response: The bacterial and gene abbreviations have been checked and full genus name added the first time it is mentioned.

The reviewer stated:  “Finally, I am grateful for the chance to review this submission. Although the topic is valuable and interesting, further interpretation and motivation is required. Wishing the authors success for their paper publication, I hope the suggestions are beneficial.”

Response: We thank the reviewer for his/her time and suggestions.

Reviewer 3 Report

This study discribed the genetic determinants that contribute to A. baumannii serum-associated AEMR by screening a transposon mutant library for members that display a loss of AEMR phenotype and find that mutation of a putative pirin-like protein, YhaK, results in a loss of AEMR, a phenotype that could be complemented by a wild type copy of the yhaK gene and was verified in a second strain background.

However, the discussion part is poor hence I recommend comparing / adding more previous studies that support this work.

Authors need to use the same formatting throughout the manuscript. Please check Page no 6, line no 272-273.

----------------

Minor editing of English language required.

Author Response

The reviewer stated:  the discussion part is poor hence I recommend comparing / adding more previous studies that support this work.

Response:  As noted above (Reviewer 2), very few laboratories have studied the effects of serum on Acinetobacter AEMR or bacterial pirins.  In fact, a literature search using the search terms “Acinetobacter” and “pirin” renders zero publication entries.  Similarly, studies of pirin-like proteins are also very few, indeed a literature search using the generic search term “pirin” yields only 96 publications- most of which are completely irrelevant.  The hand full of relevant pirin publications are largely confined to human and Escherichia coli pirins, and were discussed in the initial submission.  Furthermore, based on that discussion we have performed an amino acid alignment to hypothesize/discuss the mechanism by which YhaK may modulate AEMR and its potential as a new putative therapeutic target.  In regard to the latter, and in response to Reviewer 2, we have now included text to discuss the next steps that may be taken to further characterize the protein’s promise as a therapeutic target (lines 701-703 of revised text).  Moreover, we have added a paragraph to detail that a limitation of the current study is that transposon mutants do not allow study of essential gene products.  Further, we discuss the use of YhaK as a biochemical and/or genetic tool to identify such essential gene products (lines 707-717 of revised text).

Reviewer 4 Report

In the present study author identify genetic determinants that contribute to A. baumannii serum-associated AEMR by screening a transposon mutant library for members that display a loss of AEMR phenotype.They revealed that mutation of a putative pirin-like protein, YhaK, results in a loss of AEMR, a phenotype that could be complemented by a wild type copy of the yhaK gene and was verified in a second strain background. I accept this manuscript in the antibiotics. However, minor alternation required in the figures X axis number should not be tilted. 

1. There is 0.05 number in figure 5. no idea what is it?

2. References should be identical there is no uniformity in the references. 

Author Response

Reviewer 4:

The reviewer stated: “I accept this manuscript in the antibiotics. However, minor alternation required in the figures X axis number should not be tilted.” 

The reviewer stated: There is 0.05 number in figure 5. no idea what is it?

Response: The number 0.05 in figure 5 was a typo from a shift in the text. This has been fixed.

The reviewer stated:  References should be identical there is no uniformity in the references. 

Response: The references have been double checked and corrected.

Reviewer 5 Report

The researchers conducted a study to investigate how a specific genetic mutation called YhaK affects Acinetobacter baumannii's ability to develop serum-inducible adaptive efflux-mediated antibiotic resistance. Their goal was to evaluate whether this mutation influences the overall efflux properties of bacterial cells and contributes to antibiotic resistance. By doing so, they aimed to gain a more definitive understanding of the role of YhaK in A. baumannii's mechanisms of resistance. The manuscript presents a logical and well-designed analysis.

Minor concerns:

1. In Figure 1A, the title of the x-axis is not fully visible.

2. In lines 387-391 and 400-404, it would be beneficial if the authors also provided an analysis of the other four mutants, explaining their potential impact on the AEMR response and the reasons for excluding them.

Author Response

Reviewer 5:

The reviewer stated:  The manuscript presents a logical and well-designed analysis.  But did have some editing suggestions, as follows.

The reviewer stated:   In Figure 1A, the title of the x-axis is not fully visible.

Response: Figure 1A has been adjusted

The reviewer stated:  In lines 387-391 and 400-404, it would be beneficial if the authors also provided an analysis of the other four mutants, explaining their potential impact on the AEMR response and the reasons for excluding them.

Response: The results section has been updated to more clearly articulate that YhaK is a hypothesized regulatory protein and as such, may have a central role in AEMR.   In newly added text (Discussion) we point out that the contributions of the other gene products identified in this study are less clear (i.e. annotated as hypothetical protein).  None the less, we are currently characterizing the other proteins.

Round 2

Reviewer 2 Report

#2 I appreciate your clarification on the introduction section and the objectives of your study, I still believe that the introduction could benefit from some improvement. I would like to suggest that you revise the introduction by focusing on the specific objectives of your study and ensuring that it is clear and concise.

#4 Please consider moving all statistical analysis to the last method, rather than including them separately in each method. This will help streamline the presentation and make it easier for readers to understand the statistical approaches used in the study.

#5 To ensure that the Results section is clear and concise, I strongly suggest that you revise it to focus solely on presenting the findings of your study. Any other information and citations should be removed from the Results section.  

#8 While I am glad to hear that the bacterial and gene abbreviations have been checked and the full genus name has been added the first time it's mentioned, I noticed that there are still instances where the full scientific name of bacterial species is repeated instead of being abbreviated on subsequent mentions. For example, Acinetobacter baumannii is mentioned multiple times in Lines 141 and 191, Klebsiella pneumonia in Line 661, and Pseudomonas aeruginosa in Lines 645 and 661, among others, without using the abbreviated form of the genus. Please review the manuscript carefully and make the necessary corrections to ensure consistency in the use of abbreviations and italicization of scientific bacterial names.

Author Response

We have addressed, to the best of our ability, each of the reviewer’s recommendations.

#2 I appreciate your clarification on the introduction section and the objectives of your study, I still believe that the introduction could benefit from some improvement. I would like to suggest that you revise the introduction by focusing on the specific objectives of your study and ensuring that it is clear and concise.

Response: We have removed lines 71 to 88 of the text to hopefully provide a more concise introduction.  Further, we have edited line 109 of the revised text to explicitly state “…the objective of the current study was to identify the genetic determinants that contribute to A. baumannii AEMR”  With these changes we honestly cannot conceive how the introduction could become any more concise or the main objective of the research could be more clearly articulated. 

#4 Please consider moving all statistical analysis to the last method, rather than including them separately in each method. This will help streamline the presentation and make it easier for readers to understand the statistical approaches used in the study.

Response: We have now included a separate section within the Materials and Methods section entitled “Statistics” (lines 333-338 of revised text).

#5 To ensure that the Results section is clear and concise, I strongly suggest that you revise it to focus solely on presenting the findings of your study. Any other information and citations should be removed from the Results section. 

Response: The results section must allow the reader to understand why experiments were performed (which is often based on previous studies/references are required).  For instance, our work revealed 5 loci to be potentially important for AEMR study.  We focused on a pirin-like protein because pirins have been reported to exhibit regulatory functions- a reference is required to substantiate and guide the reader to pre-exisiting work describing pirin’s as regulators.  Similarly, we measure AEMR using the antibiotic minocycline because it is a frontrunner antibiotic for A. baumannii treatment- a reference is required to substantiate.  Moreover, AEMR occurs at levels that directly correlate with minocycline levels within humans- a reference is required to confirm.  Thus, removal of references from the results section (former) is not appropriate.  We could understand the reviewer’s concern if the manuscript was descriptive in nature (such as describing the genetic composition of a strain-set) but not for an exploratory manuscript such as ours in which experiments (and their interpretation) are predicated by literature.    

None the less, we do understand where the reviewer is coming from as technically there is “Discussion” within our results section (former).  As a work around, and in direct fitting with the instructions for authors, which states “Authors may choose to have Results and Discussions as one or two sections.”, we have elected to combine the Results with the Discussion section into a single “Results and Discussion” section.  Doing so should remove the reviewer’s concern.  Of note, we have heavily edited lines 627-638 of the discussion section (former) so that the text flows.

#8 While I am glad to hear that the bacterial and gene abbreviations have been checked and the full genus name has been added the first time it's mentioned, I noticed that there are still instances where the full scientific name of bacterial species is repeated instead of being abbreviated on subsequent mentions. For example, Acinetobacter baumannii is mentioned multiple times in Lines 141 and 191, Klebsiella pneumonia in Line 661, and Pseudomonas aeruginosa in Lines 645 and 661, among others, without using the abbreviated form of the genus. Please review the manuscript carefully and make the necessary corrections to ensure consistency in the use of abbreviations and italicization of scientific bacterial names

We have abbreviated A. baumannii in lines 36, 53, 65, 131, and 181 of the revised manuscript and P. aeruginosa in line 65 of the revised manuscript.  We found no other occurrences of Acinetobacter or Pseudomonas within the text and Klebsiella is only in the text once (line 65).